# Burnout in French General Practitioners: A Nationwide Prospective Study

**DOI:** 10.3390/ijerph182212044

**Published:** 2021-11-16

**Authors:** Frédéric Dutheil, Lenise M. Parreira, Julia Eismann, François-Xavier Lesage, David Balayssac, Céline Lambert, Maëlys Clinchamps, Denis Pezet, Bruno Pereira, Bertrand Le Roy

**Affiliations:** 1CNRS, LaPSCo, Physiological and Psychosocial Stress, University Clermont Auvergne, 63000 Clermont-Ferrand, France; fdutheil@chu-clermontferrand.fr; 2Preventive and Occupational Medicine, University Hospital of Clermont-Ferrand, 63000 Clermont-Ferrand, France; lparreira@chu-clermontferrand.fr; 3WittyFit, 63000 Clermont-Ferrand, France; 4Faculty of Medicine, Department of General Practitioner, University Clermont Auvergne, 63000 Clermont-Ferrand, France; julia.eismann@gmail.com; 5Laboratory Epsylon, Dynamic of Human Abilities & Health Behaviors, University of Montpellier, 34000 Montpellier, France; fx-lesage@chu-montpellier.fr; 6Occupational and Preventive Medicine, University Hospital of Montpellier, 34000 Montpellier, France; 7INSERM U1107 NEURO-DOL, University Clermont Auvergne, 63000 Clermont-Ferrand, France; dbalayssac@chu-clermontferrand.fr; 8Clinical Research and Innovation Direction, University Hospital of Clermont-Ferrand, 63000 Clermont-Ferrand, France; 9Biostatistics Unit, Clinical Research and Innovation Direction, University Hospital of Clermont-Ferrand, 63000 Clermont-Ferrand, France; clambert@chu-clermontferrand.fr (C.L.); bpereira@chu-clermontferrand.fr (B.P.); 10INSERM U1071, M2iSH, University Clermont Auvergne, USC-INRA 2018, 63000 Clermont-Ferrand, France; dpezet@chu-clermontferrand.fr; 11Department of Digestive Surgery, University Hospital of Clermont-Ferrand, 63000 Clermont-Ferrand, France; 12Department of General Surgery, University Hospital of Saint-Etienne, 42270 Saint-Priest-en-Jarez, France; leroybertrand8@gmail.com

**Keywords:** mental health, burnout, general practitioners

## Abstract

Background: We aimed to evaluate the prevalence of burnout among French general practitioners in private practice and to study the risk and protective factors of burnout. Methods: A nationwide cross-sectional study was conducted with French GPs working in a private practice in France who were asked to fulfil an internet questionnaire. We used the secure internet application REDCap^®^. Exclusion criteria were only working in a hospital, substitute doctors, and internship students. There was a putative sample size of 88,886 GPs. We retrieved the Maslach Burnout Inventory (MBI), occupational characteristics (type of installation, emergency regulated shifts, night shifts, university supervisor, weekly hours worked, seniority), and personal characteristics such as age, gender, marital status, and number of children. Results: We included 1926 GPs among the 2602 retrieved questionnaires. A total of 44.8% of French liberal GPs were experiencing burnout, with 4.8% (95%CI 3.9–5.9%) experiencing severe burnout. The risk factors of severe burnout were male gender (RR = 1.91, 95%CI 1.15–3.16), working in a suburban area (5.23, 2.18–12.58), and having more than 28 appointments per day (1.95, 1.19–3.19). Working more than 50 h weekly showed a tendency to increase the risk of severe burnout (1.55, 0.93–2.59, *p* = 0.095), with a significant increase in the risk of low and moderate burnout (1.31, 1.02–1.67 and 1.86, 1.34–2.57, respectively). Protective factors were mainly resident training, which decreased the risk of both low, moderate, and severe burnout (0.65, 0.51–0.83; 0.66, 0.48–0.92; and 0.42, 95%CI 0.23–0.76, respectively). Performing home visits decreased the risk of severe burnout (0.25, 0.13–0.47), as did group practice for intermediate level of burnout (0.71, 0.51–0.96). Conclusion: GPs are at a high risk of burnout, with nearly half of them in burnout, with burnout predominantly affecting males and those between the ages of 50 and 60 years old. The main risk factors were a high workload with more than 28 appointments per day or 50 h of work per week, and the main protective factors were related to social cohesion such having a teaching role and working in a group practice with back-office support.

## 1. Introduction

Several nationwide studies have demonstrated that General Practitioners (GPs) are particularly at risk of burnout. The high prevalence of burnout seems to be global in GPs, as demonstrated through nationwide studies from Europe [1,2,3,4], America [5], Asia [6,7], and Africa [8]. However, nationwide studies with a good representation of French GPs are lacking. Moreover, GP practice in France has some specificities, such as GPs total freedom in their mode of practice. Studying factors influencing burnout are of particular interest to build preventive effective strategies and to avoid its public health consequences. In addition to sociodemographic such as age [9,10,11], gender [10,11,12], and family status [12], some occupational characteristics are also known to be risk factors of burnout such as workload [11,13] and long work duration [12]. Similarly, GPs are also exposed to emergencies during night shifts, which can be particularly stressful [14,15,16] and can be linked with burnout [17]. However, there are some protective factors for burnout, such as working in a healthy environment [18,19] and social cohesion [20]. For GPs, social cohesion can be related to more clinical interaction [18,21] and less administrative tasks [11,12]. Despite being sometimes viewed as being time consuming, home visits could also represent a sense of engagement and a clinically rewarding sense [22]. Particularly, in France, there is also a framework that allows GPs to practice either in a solo or group cabinet [11,12,13,20] and to tutor medical students and interns [9,20], which can fuel a sense of social cohesion. Moreover, no studies quantified the risk of burnout while combining all of those risk and protective factors. European studies show that GPs in rural areas seem to be more affected by burnout symptoms than GPs in urban regions are [9,11,12,20,21].

Therefore, the aim of this study was to evaluate the prevalence of burnout among French general practitioners in private practice and to determine the risk and protective factors of burnout.

## 2. Methods

### 2.1. Study Design

This nationwide cross-sectional study took place over six consecutive months before the start of the COVID-19 pandemic.

Volunteer GPs were asked to fulfil an internet questionnaire. The questionnaire was created in a multidisciplinary way (doctors, epidemiologists, biostatisticians, etc.) and was tested on ten doctors before generalized sending. The questionnaires were completed anonymously. The online questionnaire was diffused several times through the mailing lists of the National, Regional and Departmental Physicians Council, the Regional Union of healthcare workers (URPS) and was published in two social groups for physicians on social media (Facebook and Linkedin). We used the secure internet application REDCap^®^ to build and manage the questionnaire, which was hosted by the University Hospital of Clermont-Ferrand. The study was reviewed and approved by the human ethics committees Sud-Est VI (Clinicaltrials.gov (accessed on 8 October 2021) NCT04242862).

### 2.2. Participants

To be included, the participants had to be general practitioners in a private practice in France. The exclusion criteria were only working in a hospital, substitute doctors, and internship students. According to the figures from the French Medical Council (CNOM), there was a putative sample size of 88,886 GPs in France (Figure 1).

### 2.3. Instrument Survey—Outcomes

#### 2.3.1. Burnout

The Maslach Burnout Inventory (MBI) has demonstrated its external validity among GPs [23] and has been validated in French [24]. The MBI is composed of 22 items designed to assess the three dimensions of burnout syndrome: emotional exhaustion (9 items), depersonalization (5 items), and reduced personal accomplishment (8 items). The items are written in the form of statements concerning personal feelings or attitudes. Items are composed of a 7-point scale determining the frequency of feelings, with items varying from “never” to “every day”. The scores for each dimension of burnout syndrome are considered separately and are not combined into a single total score. Scores for each dimension can be coded as low, average, or high. We considered that there was no burnout when all of the three dimensions report a low risk, low burnout when there was a high risk of burnout in one of the three dimensions, intermediate burnout when there was a high risk of burnout in two dimensions, and severe burnout when there was a high risk of burnout in the three dimensions [25].

#### 2.3.2. Occupational Characteristics

We assessed career characteristics (career length, i.e., seniority within profession, and further qualifications such as specific complementary diplomas), setting characteristics (setting location—rural, suburban, or urban— and commuting time), social environment (group practice, home visits, and resident training as a university supervisor), work support (administrative support and partly salaried), workload (number of hours worked per week, and number of daily consultations), and emergency activity (yes/no, realisation of night emergency-regulated shifts in private practice, and night shifts in an emergency hospital department).

#### 2.3.3. Personal Characteristics

Age, gender identity, marital status, and whether the participants had children were assessed.

### 2.4. Statistics

We estimated that the sample size should comprise at least 1536 individuals using an expected prevalence of severe burnout ranging between 10% and 20% from the EGPRN study [26], allowing us to highlight results with a 2% error margin for a 95% confidence level. All of the statistical analyses were performed with Stata^®^ software (version 15, StataCorp, College Station, TX, USA). The categorical variables were described using frequencies and percentages, whereas the continuous variables were presented with mean ± standard deviation or median (interquartile range) according to their statistical distribution. The normality was accessed using the Shapiro-Wilk test. Confidence intervals (95%CI) for burnout prevalence according to the MBI were estimated using the exact binomial approach. Then, the comparisons of the burnout prevalence between groups were performed using the Chi2 or Fisher’s exact tests for the categorical variables and by using ANOVA or the Kruskal-Wallis tests for quantitative variables. The assumption of homoscedasticity was analysed with Bartlett’s test. When appropriate (omnibus *p*-value was less than 0.05), post hoc tests for two-by-two multiple comparisons were applied: Tukey-Kramer after ANOVA, Dunn after Kruskal-Wallis, and Marascuilo for categorical variables. The relationships between the quantitative variables were analysed using the Pearson or Spearman correlation coefficients by applying Sidak’s type I error correction. Finally, multivariate analysis was conducted with covariates chosen according to their clinical relevance and to the univariate results. More precisely, ordinal polynomial regression (i.e., ordered logit models of dependent ordinal variable) was conducted. Particular attention was paid to the study of multicollinearity and interactions between covariates (1) studying the relationships between the covariates and (2) evaluating the impact of adding or deleting variables on a multivariate model. The results were expressed as relative-risks and 95% confidence intervals (95%CI), and forest plots were employed to present the results. A Sidak’s type I error correction was applied to take into account multiple comparisons. Multivariate analysis was performed for 1802 patients (i.e., 124 (6.4%) missing data). A sensitivity analysis was conducted to analyse the statistical nature of the missing data. A two-sided type I error of 5% was applied for all statistical tests. As such, a difference was considered statistically significant when *p* ≤ 0.05.

## 3. Results

We received 2602 questionnaires, with 1926 GPs matching the inclusion criteria, representing a response rate of 3.6% of the 56430 national GP population meeting the criteria (Figure 1).

### 3.1. Burnout Prevalence

According to the MBI, 44.8% of the French liberal GPs were experiencing burnout: 4.8% (95%CI 3.9–5.9%) had severe burnout (severe scores within the 3 dimensions), 13.9% (95%CI 12.4–15.5%) had intermediate burnout (scoring severe within two dimensions), and a quarter of them (26.1%, 95%CI 24.1–28.1%) had a low level of burnout (severe scores within one dimension). About half had no severe burnout scores (55.2%, 95%CI 52.9–57.4%), regardless of the dimension (Table 1).

### 3.2. Burnout by Sociodemographic and Occupational Characteristics

#### 3.2.1. Sociodemographic Variables

From the 1926 samples that were obtained, about half were men (52.3%, *n* = 1004) (Figure 2). The mean global age was 50.0 ± 10.7 years, with the men being 8 years older than the women were on average (53.9 ± 9.6 vs. 45.7 ± 10.1, *p* < 0.001). Both male GPs and those over the age of 50 years old were significantly more affected by severe burnout (6.2% vs. 3.4%, *p* < 0.001 and 5.7% vs. 3.6%, *p* = 0.033). Most of the respondents were married or in a couple (85.6%, *n* = 1644) and had children (89.0%, *n* = 1709), but none of those personal particularities influenced their risk of severe burnout. In addition to all of those risk factors of severe burnout, working more than 50 h per week was also a risk factor for low and moderate burnout.

#### 3.2.2. Career Characteristics

More than half of the respondents had worked for more than 20 years (57.4%, *n* = 1087) and had acquired a supplementary qualification (56.2%, *n* = 1219), with male doctors having almost 10 more years of clinical experience (24.8 ± 10.4 vs. 16.3 ± 10.3, *p* < 0.001) and a slightly higher prevalence of further qualifications than women (48.1% vs. 43.8, *p* = 0.063). Career length longer than 20 years almost emerged as a factor associated with severe burnout (*p* = 0.059). Benefiting from further qualifications was neither a risk factor nor a protective factor for burnout (*p* = 0.793). Results were similar through all burnout levels.

#### 3.2.3. Setting Characteristics

Almost the same proportion of participants worked in urban or in suburban areas (38.8% and 39.9%, respectively), and only around 20% worked in rural areas (mostly males). Suburban physicians had a higher severe burnout rate than their rural or urban colleagues (7.1% vs. 3.4%, *p* < 0.001). The mean time spent during the journey from home to work was 12.0 ± 11.3 min and was the same across burnout levels (*p* = 0.527).

#### 3.2.4. Social Practice Context

The majority worked with other doctors in a group practice (65.4%, *n* = 1241), performed home visits (91.6%, *n* = 1740), and were invested in training a resident doctor (31.8%, *n* = 608), with a higher rate among rural practicians (25.6% vs. 27.9%, *p* < 0.001). All of those factors were related to a lower prevalence of severe burnout: 3.6% vs. 6.7% whether working in group or alone, 4.1% vs. 11.3% whether performing home visits or not, 2.8% vs. 5.9% whether training residents or not (*p* < 0.001).

#### 3.2.5. Work Support

The GPs who had a secretariat (76.5%, *n* = 1436) showed a lower burnout rate (74.6% vs. 78.1%, *p* = 0.050), although working onsite or at home had no significant effect. Being an employee (14.8%, *n* = 283) was not related to the risk of burnout (*p* = 0.755).

#### 3.2.6. Workload

The GPs worked 50.5 ± 12.4 h per week on average. The nearly 60% of the participants exceeding that amount had more severe burnout (5.7% vs. 3.6%, *p* < 0.001); GPs with severe burnout also worked around 5 h more than those who were not experiencing burnout (53.7 ± 12.9 vs. 49.1 ± 12.1 h, *p* < 0.001). GPs seeing more than the average of 28.5 ± 11.1 consultations per day had more severe burnout (64 vs. 48%, *p* = 0.003). Men worked 8 h more than their women counterparts (54.3 ± 12.3 vs. 46.4 ± 11.0, *p* < 0.001) and saw about five more patients per day (30.9 ± 12.6 vs. 26.2 ± 8.4, *p* < 0.001).

#### 3.2.7. Emergency Activity

Among the 60.9% (*n* = 1166) of GPs who reported having emergency activity, 89.0% were regulated by the Emergency Medical Dispatch (EMD) (in private practice), and 18.9% performed night shifts (in emergency departments). Performing shifts did not appear to be a risk factor for burnout (*p* = 0.673), whether they were regulated or not (*p* = 0.650).

### 3.3. Relationships between Other Variables Than Burnout

#### 3.3.1. Sociodemographic Variables

The male GPs were statistically older (53.9 ± 9.6 vs. 45.7 ± 10.1 for women, *p* < 0.001), had a longer career (24.8 ± 10.4 vs. 16.3 ± 10.3, *p* < 0.001), were more likely to practice in group, have a secretariat, worked more hours, saw more patients, and had a lower commuting time (*p* < 0.001). On average, those who had children spent had a commute from home that was approximately 5 min shorter (*p* < 0.001).

#### 3.3.2. Setting Characteristics

The suburban doctors had an overall similar workload to their urban colleagues, but and were more frequently associated with social practice characteristics, which were stated either as working in group practice, having a secretariat, or as performing home visits. A rural practice implied higher weekly working hours, daily appointment number, and more frequent resident training than the other settings, but it negatively correlated with emergency activities at night.

#### 3.3.3. Social Practice Context

GPs completing home visiting worked more weekly hours than those who did not (50.8 ± 12.0 vs. 46.6 ± 15.1, *p* < 0.001) while keeping the same daily number of patients (28.6 ± 10.8 vs. 28.3 ± 14.2, *p* = 0.732). Almost all of the rural and suburban doctors (respectively 98.1% and 95.3%) performed home visits. In comparison, urban doctors performed less home visits than other groups (84.1%, *p* < 0.001). Less than a third were invested in training a resident doctor (31.8%, *n* = 608), with similar rates across the different geographical areas (urban: 30.9%; suburban: 37.6%; rural: 31.6%). Intern tutoring was not associated with any geographical setup. All of those protective factors were more associated with suburban activity and less with rural activity. Regardless of the geographical area, the proportion of GPs with residents was the same. The doctors across different areas undertook a similar workload, which was evaluated either through weekly hours (49.8 ± 12.4 vs. 50.5 ± 12.4) or through daily consultations (29.4 ± 12.0 vs. 28.6 ± 10) (not significant).

#### 3.3.4. Work Support, Workload, and Emergency Activity

GPs with a secretariat worked more frequently in a group practice (87.8% vs. 12.2%, *p* < 0.001), and worked less weekly hours (*p* = 0.04). Emergency activity, especially during the night was more frequent for males.

### 3.4. Multivariate Analysis

The risk factors of severe burnout were male gender (RR = 1.91, 95%CI 1.15–3.16), working in a suburban area (5.23, 2.18–12.58), and having more than 28 appointments per day (1.95, 1.19–3.19) (Figure 3). Working more than 50 h weekly showed a tendency to increase the risk of severe burnout (1.55, 0.93–2.59, *p* = 0.095), with a significant increase in the risk of low and moderate burnout (1.31, 1.02–1.67 and 1.86, 1.34–2.57, respectively). Protective factors were mainly resident training which decreased the risk of both low, moderate, and severe burnout (0.65, 0.51–0.83; 0.66, 0.48–0.92; and 0.42, 0.23–0.76, respectively). Performing home visits decreased the risk of severe burnout (0.25, 0.13–0.47), as did group practice for the intermediate level of burnout (0.71, 0.51–0.96).

## 4. Discussion

The main findings of the present study are that we demonstrated a massive prevalence of burnout in GPs, with nearly half of them with some degree of burnout, predominantly affecting males and those between 50 and 60 years old. The main risks factors were working in suburban area, a high workload, which was defined as having more than 28 appointments per day, and male gender. The main protective factors were related to social cohesion, such having a teaching role and conducting home visits.

### 4.1. Burnout Prevalence

In our study, half of the GPs had some degree of burnout and were experiencing it in similar level of prevalence described in the literature [1,2,3,4,5,6,7,8,26,27,28]. Nevertheless, only 4.8% French liberal GPs were experiencing severe burnout, a much lower prevalence than those previously reported in similar nationwide studies [27,28]. This difference may be due to the French legal framework in which GPs operate. Indeed, GPs have the flexibility to organize their practice (work with or without a secretariat, interns, other colleagues, emergency activities, home visiting habits). Thus, they can organize their practice regarding determinants of burnout, such as workload, job control, effort rewards, community, fairness, and values [13,20,25,26,27,28]. Applying the framework of French legal specificities could reduce the prevalence of severe burnout among GPs in others countries [29]. A recent meta-analysis on burnout of French physicians that included 37 studies reported a similar prevalence of burnout (48%), with 5% experiencing severe burnout [30]. No statistical difference has been found between other medical specialties, with the exception of emergency physicians, with 57% of them experiencing burnout and 12% experiencing severe burnout [30]. The mean prevalence of burnout in physicians is higher than in non-medical professions [31]. However, meta-analysis on nurses including, 113 publications showed a high prevalence of severe burnout (11.2%) [32]. Interestingly, the burnout rate in physicians seemed to increase with time [33,34], whereas a meta-analysis showed that suicide rate of physicians decreased over time, especially in Europe [35]. Burnout may also negatively affect patient care by increasing medical errors and decreasing productivity and empathy toward patients [36]. Physicians with burnout are twice as likely to deliver unsafe care, have unprofessional behavior, and cause patient unsatisfaction [37]. Burnout is also associated with a large number of physical, psychological, occupational, and social consequences, such as cardiovascular disease, musculoskeletal pain, alcohol consumption, obesity, sleep disorders, or dehumanization [38].

### 4.2. Risk Factors

We demonstrated that the main risk factors of burnout were to be a man, working in semi-rural areas, and have a high workload. As previously described in other studies [21,28], the risk of severe burnout in men was three times higher than in women. Men combined several risk factors, such as accepting high workloads [39] and working alone more often, without other GPs or a secretariat. In France, contrary to other European studies [9,10], older GPs had more burnout than younger ones. Previous GPs generations admitted to working closer to their limits [39] and thought of an early retirement [40]. Working nearby older colleagues experiencing burnout can discourage the recruitment and installation of young GPs [41]. The expectations that young doctors have about family life and work–life balance [42] might lead them to abandon the specialty [39,41,42]. Although not statistically different, single GPs present a higher burnout rate against those in a couple. Previous studies showed that marital status had a similar effect on on the burnout [39]. As expected, high burnout was more likely with a higher workload [27] but not among those who had worked during emergency activities, unlike what has been described in the literature [27,28]. As the amount of revenue that GPs make is tied to their number of consultations, they are prone to accept and deal with more patients [43,44]. In the literature, administrative tasks are also commonly described as a burden by GPs [43,45]. Transferring duties to staff members or having a secretariat increases office efficiency, patient satisfaction, productivity, and well-being [46,47]. Being a part-time employee in a health centre also allows social cohesion, less administrative tasks, and provides a fixed salary and social benefits [48]. However, it imposes the constraints of working in an enterprise, i.e., lower autonomy, imposed cadence and working methods [49], shared workspaces [12,20] and budget constraints [13,44].

### 4.3. Protective Factors

In our study, the main protective factors were those bringing a social cohesion [41], such as working in a group practice, having a teaching role, or the geographical area of practice. Group practice has also been previously described as a protective factor of burnout [48]—even if some studies did not find any influence [12,27,50]. Although some GPs do not usually seek their peers for advice—alleging lack of time—they recognize that clinical discussions solved during clinical discussions [21]. Encouraging peer arrangements should be a key intervention [21,41,51], both within the specialty and with other specialists [29,41]. Group practice may also benefit the socials needs that physicians have. Similarly, resident training has been consistently associated with a GPs experiencing a lower burnout rate, as it seems to promote regular clinical interactions and therefore avoid isolation [27]. Through the distribution of tasks to their residents [21], they may also reduce their workload and may even improve their own technical skills [21,41]. For solo practitioners, resident training may also reduce the perception of isolation. Lastly, the geographical area of the practice influenced burnout in our study. In line with the literature [28,50], working in rural areas is a protective factor of burnout. Even though national rural areas have difficulties in recruiting doctors, they had a significantly lower burnout prevalence. This may be related to the greater satisfaction given by their relationship with patients, clinical autonomy, and life in small communities [2,12,27,49,51,52,53,54].

### 4.4. Limitations and Strengths

All of the limitations of the present study are followed by counterbalanced strengths. The main limitation is a cross-sectional study design; however, the primary aim was to report the prevalence of burnout in GPs, and this design allowed us to work with a large number of responses. Some councils and unions submitted less responses, which may have weakened geographical representativeness. The generalizability of our results may also be precluded by the high proportion of male respondents as well as general practitioners who are older than 50 years old. Furthermore, the French specificities of the practice of general medicine could prevent a generalization of results in other countries. Nevertheless, this is the first French nationwide study on GP burnout. We obtained a large sample size of respondents, allowing us to be confident in the accuracy of our data and giving us enough statistical power to study influencing factors with multivariate analysis. Our study is also one of the highest collections of burnout-related data in GPs compared to studies in other countries. It could have been interesting to compare the characteristics of the respondents and non-respondents; however, data were not available for the non-respondents, and other studies on GP burnout also lacked those comparisons. GPs prone to burnout might have been more prone to answer the questionnaire, inducing a selection bias, but this putative bias is common to any questionnaire—including all previous questionnaires concerning the prevalence of burnout among GPs from other countries [1,2,3,4,5,6,7,8]. The length of the questionnaire, due to the MBI scale and to diversity of factors explored, might have caused some GPs to refrain from responding. However, the MBI is a well-proven instrument, making comparisons with other studies possible. The comprehensive questionnaire also allowed us to retrieve data on our secondary outcomes, i.e., variables associated with burnout according to the literature. However, to guarantee the feasibility of a study with a short and relevant questionnaire, some missing variables that may have played an important role, have not been collected. For example, the work choice of GPs and their burnout rate may be related to income pressure, but this topic was not included. Our inclusion criteria were very strict, yet we had a homogeneous population of liberal GPs. The collection period of 6 months allowed a real screenshot of burnout at a specific time. Finally, it would be interesting to reproduce the study after the COVID-19 pandemic to evaluate the impact of the coronavirus pandemic on this front-line population [55,56,57].

## 5. Conclusions

General practitioners are at high risk of burnout, with nearly half of them experiencing burnout, predominantly affecting, especially those between 50 and 60 years old. The main risk factors for burnout were a high workload, and the main protective factors were related to social cohesion, such as having a teaching role and working in a group practice with back-office support. Preventive strategies should be implemented to improve the well-being of general practitioners.

## Figures and Tables

**Figure 1 ijerph-18-12044-f001:**
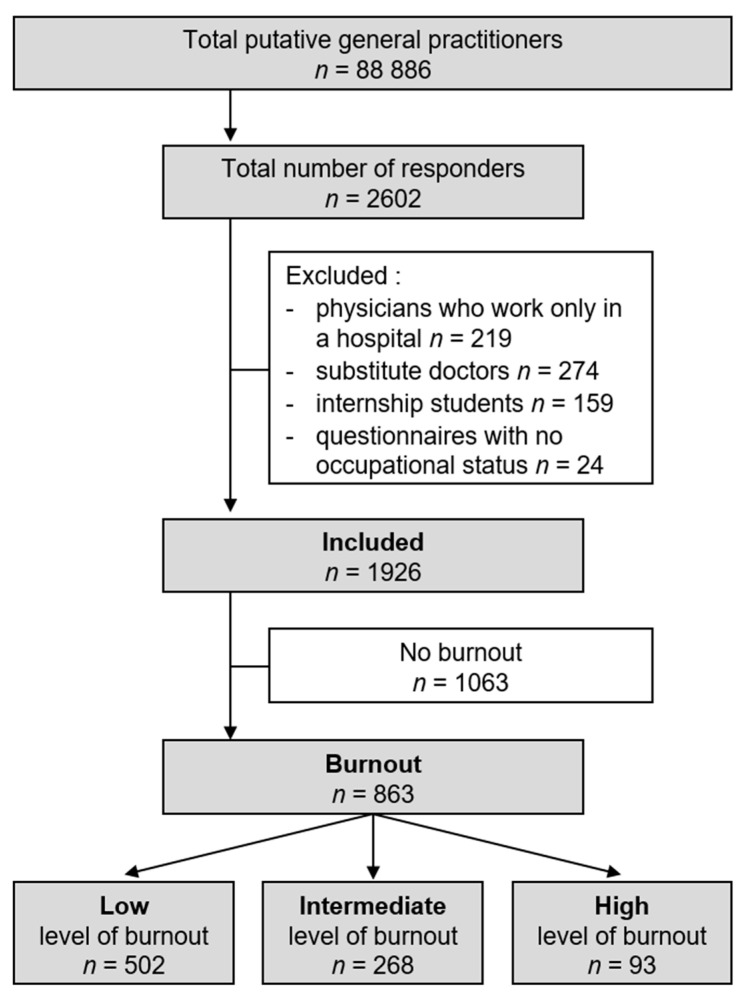
Flow chart.

**Figure 2 ijerph-18-12044-f002:**
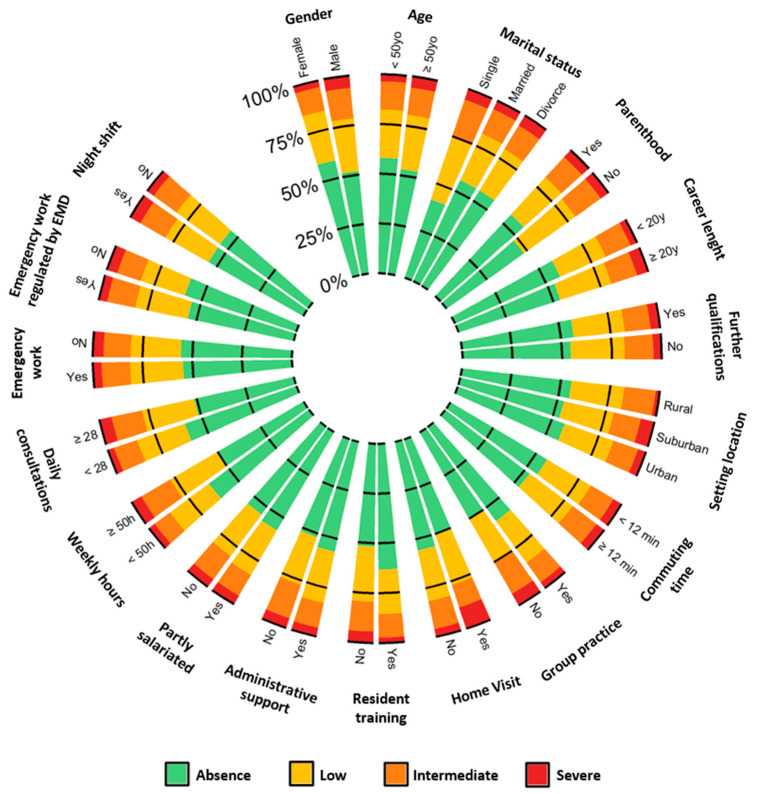
Polar histogram of levels of burnout depending on variables (see Appendix A for details of percentages).

**Figure 3 ijerph-18-12044-f003:**
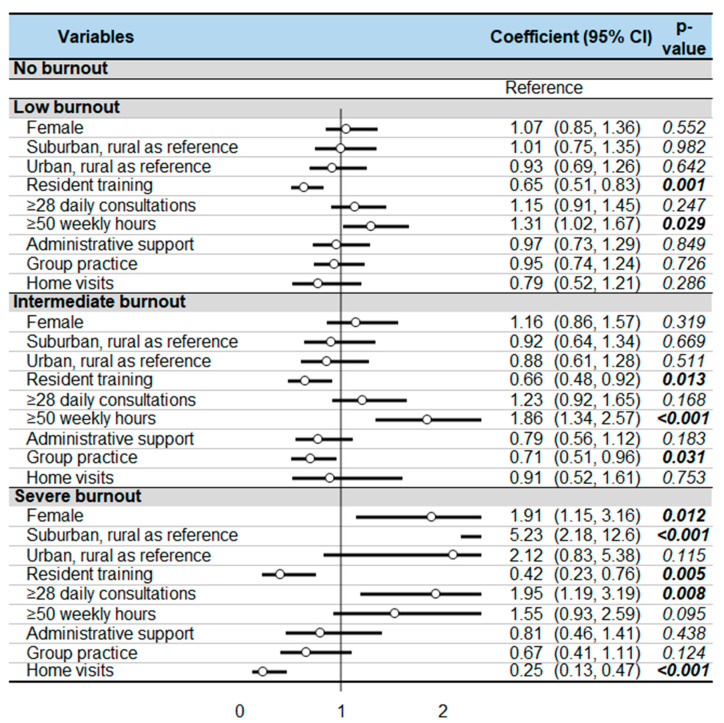
Multivariate analysis with results expressed as relative risks (and 95%CI). Bold *p*-value for significant risk or protective factor.

**Table 1 ijerph-18-12044-t001:** Burnout by sociodemographic and occupational characteristics.

	Total Sample	Level of Burnout				*p*-Value
		No Burnout	Low Burnout	Intermediate Burnout	Severe Burnout	
	*n* (%) or Mean ± SD	*n* (%) or Mean ± SD	*n* (%) or Mean ± SD	*n* (%) or Mean ± SD	*n* (%) or Mean ± SD	
Total	1926 (100%)	1063 (55.2%)	502 (26.1%)	268 (13.9%)	93 (4.8%)	**<0.001**
Sociodemographic						
Gender						
Female	916 (47.7%)	540 (50.9%)	234 (46.7%)	111 (41.9%)	31 (33.3%)	**<0.001**
Male	1004 (52.3%)	521 (49.1%)	267 (53.3%)	154 (58.1%)	62 (66.7%)	
Age						
mean	50.0 ± 10.7	49.7 ± 11.2	50.2 ± 10.5	49.8 ± 9.8	52.5 ± 8.6	0.098
<50 years old	840 (43.8%)	488 (46%)	204 (40.7%)	118 (44.4%)	30 (32.6%)	**0.033**
≥50 years old	1080 (56.2%)	573 (54%)	297 (59.3%)	148 (55.6%)	62 (67.4%)	
Marital status						
Single	89 (4.6%)	37 (3.5%)	29 (5.8%)	18 (6.7%)	5 (5.4%)	0.214
Married or in couple	1644 (85.6%)	920 (86.6%)	425 (85.2%)	223 (83.5%)	76 (82.6%)	
Divorced, separated, or widowed	187 (9.8%)	105 (9.9%)	45 (9%)	26 (9.8%)	11 (12%)	
Parenthood						
Yes	1709 (89%)	958 (90.5%)	433 (86.6%)	235 (87.7%)	83 (89.2%)	0.125
No	211 (11%)	101 (9.5%)	67 (13.4%)	33 (12.3%)	10 (10.8%)	
Career characteristics						
Career length						
mean	20.7 ± 11.2	20.5 ± 11.6	20.9 ± 11.2	20.4 ± 10	23 ± 9.4	0.222
<20 years	807 (42.6%)	467 (44.4%)	200 (40.7%)	112 (43.1%)	28 (30.8%)	0.059
≥20 years	1087 (57.4%)	584 (55.6%)	292 (59.3%)	148 (56.9%)	63 (69.2%)	
Further qualifications						
Yes	1219 (56.2%)	665 (55.3%)	329 (57.3%)	174 (58.4%)	51 (52.6%)	0.624
No	952 (43.8%)	537 (44.7%)	245 (42.7%)	124 (41.6%)	46 (47.4%)	
Setting characteristics						
Setting location						
Rural	427 (22.3%)	238 (22.5%)	113 (22.7%)	67 (25.1%)	9 (9.7%)	**0.008**
Suburban	746 (38.9%)	398 (37.6%)	192 (38.5%)	103 (38.6%)	53 (57%)	
Urban	743 (38.8%)	422 (39.9%)	193 (38.8%)	97 (36.3%)	31 (33.3%)	
Commuting time (min)						
mean	11.8 ± 11.3	11.3 ± 10.5	12.3 ± 11.1	12.8 ± 14.1	12.9 ± 12.4	0.088
<12 min	1184 (62.2%)	668 (63.5%)	303 (60.8%)	156 (59.1%)	57 (62%)	0.527
≥12 min	722 (37.8%)	384 (36.5%)	195 (39.2%)	108 (40.9%)	35 (38%)	
Social environment						
Group practice						
Yes	1242 (65.4%)	725 (69%)	321 (65.1%)	151 (56.6%)	45 (50.6%)	**<0.001**
No	657 (34.6%)	325 (31%)	172 (34.9%)	116 (43.4%)	44 (49.4%)	
Home visits						
Yes	1740 (91.6%)	970 (92.4%)	451 (91.3%)	248 (92.9%)	71 (79.8%)	**<0.001**
No	160 (8.4%)	80 (7.6%)	43 (8.7%)	19 (7.1%)	18 (20.2%)	
Resident training						
Yes	608 (31.8%)	384 (36.4%)	136 (27.5%)	71 (26.8%)	17 (18.3%)	**<0.001**
No	1300 (68.2%)	671 (63.6%)	359 (72.5%)	194 (73.2%)	76 (81.7%)	
Work support						
Administrative support						
Yes— on site or by phone	1436 (76.5%)	813 (78%)	376 (77.2%)	183 (70.1%)	64 (73.6%)	**0.050**
No	441 (23.5%)	229 (22%)	111 (22.8%)	78 (29.9%)	23 (26.4%)	
Partly salaried						
Yes	283 (14.8%)	162 (15.3%)	73 (14.7%)	34 (12.7%)	14 (15%)	0.755
No	1632 (85.2%)	895 (84.7%)	424 (85.3%)	234 (87.3%)	79 (85%)	
Workload						
Weekly hours						
Mean	50.6 ± 12.4	49.1 ± 12	51 ± 11.9	54.4 ± 13.4	53.7 ± 13	**<0.001**
<50 h/weeks	805 (41.9%)	503 (47.4%)	196 (39.1%)	77 (29%)	29 (31.5%)	**<0.001**
≥50 h/weeks	1115 (58.1%)	558 (52.6%)	305 (60.9%)	189 (71%)	63 (68.5%)	
Number of daily consultations						
Mean	28.6 ± 11.1	27.7 ± 9.8	28.9 ± 10.3	30.6 ± 15.9	32 ± 10.5	**<0.001**
<28 consultations/day	961 (51.2%)	569 (54.7%)	243 (49.9%)	117 (45%)	32 (36%)	**<0.001**
≥28 consultations/day	915 (48.8%)	471 (45.3%)	244 (50.1%)	143 (55%)	57 (64%)	
Emergency activity						
Emergency work						
Yes	1166 (60.9%)	639 (60.4%)	309 (62.2%)	166 (61.9%)	52 (55.9%)	0.673
No	750 (39.1%)	419 (39.6%)	188 (37.8%)	102 (38.1%)	41 (44.1%)	
Emergency work regulated by EMD						
Yes	1033 (88.9%)	563 (88.1%)	278 (90.3%)	148 (90.2%)	44 (86.3%)	0.650
No	129 (11.1%)	76 (11.9%)	30 (9.7%)	16 (9.8%)	7 (13.7%)	
Night shifts						
Yes	221 (18.9%)	121 (18.9%)	55 (17.8%)	32 (19.3%)	13 (25.5%)	0.638
No	944 (81.1%)	518 (81.1%)	254 (82.2%)	134 (80.7%)	38 (74.5%)	

Bold *p*-value for significant difference.

## Data Availability

All relevant data were included in the paper.

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
