# Peer review of "Burnout in French General Practitioners: A Nationwide Prospective Study"

_ijerph, 2021, doi:10.3390/ijerph182212044_

Round 1

Reviewer 1 Report

English language editing is needed as the use of words at times are inappropriate and had to be guessed as to meaning.

Overall aim signified that there was an intervention, when there was not an intervention. The aim could be restated to only include what the study covered and not the development and testing of an intervention.

Sex as a variable was not defined. Is this assigned sex at birth or is this self-reported sex/gender identity. The latter would impact how the results are reported and analyzed.

The results section was adequate. 

Results were again reported in the discussion section and this is inappropriate and these should be removed. The discussion is for comparing and contrasting the new results to previous research and how their new results change how the topics are considered. This section will need to be redone.

Reviewer 2 Report

The paper is interesting, analyzes a quite big sample and the researchers used a fair methodology. Nevertheless, there are some concerns to be addressed, in my opinion, as follows:

  • The recruitment procedure of the participants should be described in a more detailed and appropriate way
  • The strengths of the study should be discussed (as Authors have done for the weaknesses)
  • Use "gender" instead of "sex"
  • The statistically significant level is set to "equal or less" or "less than" 0.05?
  • What about the generalizability of the results?

Round 2

Reviewer 1 Report

Revisions have been addressed appropriately.

Thank you for allowing me to review your revisions.

Reviewer 2 Report

Authors have addressed all the comments